# Tree Rings Record of Long-Term Atmospheric Hg Pollution in the Monte Amiata Mining District (Central Italy): Lessons from the Past for a Better Future

Silvia Fornasaro [1], Francesco Ciani [2], Alessia Nannoni [2], Guia Morelli [3], Valentina Rimondi [2], Pierfranco Lattanzi [3], Claudia Cocozza [4], Marco Fioravanti [4] and Pilario Costagliola [2,*]

1 Department of Earth Sciences, University of Pisa, Via Santa Maria 53, 56126 Pisa, Italy; silvia.fornasaro@unipi.it

2 Department of Earth Sciences, University of Florence, Via G. La Pira 4, 50121 Florence, Italy; francesco.ciani@unifi.it (F.C.); alessia.nannoni@unifi.it (A.N.); valentina.rimondi@unifi.it (V.R.)

3 Consiglio Nazionale delle Ricerche, Istituto di Geoscienze e Georisorse, Via G. La Pira 4, 50121 Florence, Italy; guia.morelli@igg.cnr.it (G.M.); pierfrancolattanzi@gmail.com (P.L.)

4 Department of Agricultural, Food, Environmental and Forestry Sciences and Technologies, University of Florence, Piazzale delle Cascine 18, 50144 Florence, Italy; claudia.cocozza@unifi.it (C.C.); marco.fioravanti@unifi.it (M.F.)

* Correspondence: pilario.costagliola@unifi.it

**Abstract:** Trees may represent useful long-term monitors of historical trends of atmospheric pollution due to the trace elements stored along the tree rings caused by modifications in the environment during a tree's life. Chestnut (*Castanea sativa* Mill.) tree trunk sections were used to document the yearly evolution of atmospheric Hg in the world-class mining district of Monte Amiata (MAMD; Central Italy) and were exploited until 1982. An additional source of Hg emissions in the area have been the active geothermal power plants. A marked decrease (from >200 μg/kg to <100 μg/kg) in Hg contents in heartwood tree rings is recorded, likely because of mine closure; the average contents (tens of μg/kg) in recent years remain higher than in a reference area ~150 km away from the district (average 4.6 μg/kg). Chestnut barks, recording present-day Hg pollution, systematically show higher Hg concentrations than sapwood (up to 394 μg/kg in the mining area). This study shows that tree rings may be a good record of the atmospheric Hg changes in areas affected by mining activity and geothermal plants and can be used as a low-cost biomonitoring method for impact minimization and optimal resource and land management.

**Keywords:** dendrochemistry; tree ring; *Castanea sativa* Mill.; geothermal power plant; mining activity; mercury; Monte Amiata

## 1. Introduction

A fundamental requirement of sustainable mining is the ability of predicting potential adverse environmental impacts, even in the long-term. In this study, we present an example of the reconstruction of the long-term record of mercury (Hg) variations in an area affected by Hg mining, metallurgical activities, and the exploitation of geothermal fluids. Mercury is a global and persistent contaminant which generally occurs in the atmosphere in three main species. The dominant species is gaseous elemental mercury ($Hg^0$, GEM), which comprises >95% of the global atmospheric Hg reservoir. It is a transboundary pollutant due to its low reactivity, long-range transport, and long-term residence time in the atmosphere (0.5–2 years). The other two less abundant Hg species in the atmosphere, reactive gaseous mercury ($Hg^{2+}$, RGM) and particle-bound mercury (Hgp, PBM), have much shorter residence times (hours to weeks) and are typically deposited near their Hg emission sources [1–8].

Characterizing such anthropogenic sources and quantifying Hg emissions from them are crucial objectives as they may constitute two-thirds of the present global Hg cycle, raising Hg global concentrations four to five times the background levels [9,10]. Such an increase can pose public health concern for workers and local populations through inhalation exposure [11]. Sensitive instrumentation and passive samplers (both abiotic and biotic) have been utilized to quantify Hg air concentrations and observe temporal trends [12,13]. However, these data are not representative of past trends. Data on historical trends of atmospheric Hg deposition, especially on the natural conditions preceding the pollution advent, represent an important gap in the understanding of large-scale Hg pollution and Hg global cycle [14]. At present, the application proxies for investigating Hg contamination over the long-term are lake cores, ice cores, and tree rings [14–17]. Among natural samplers, the dendrochemical analysis of tree rings proved to be an adequate tool to record atmospheric concentrations of Hg$^0$ ([8] and the references therein, [18–21]). Dendrochronological methods provide a simple, precise, and accurate means of dating with a fine temporal resolution that can be advantageous, despite the short life span of trees relative to longer geochemical archives (i.e., sediment, peat, and ice) [8,22]. On the other hand, dendrochronology is not completely accepted by the scientific community. The debate is still ongoing, because many factors influence the applicability of dendrochemical analyses, including differences in: (i) tree species and physiology (e.g., differences between sapwood and heartwood, tree age, stature, growth rate) [23–25]; (ii) Hg formation in the air; (iii) soil chemistry [26,27]; and (iv) tree location (e.g., proximity to the Hg sources, elevation, proximity to the sea/ocean, wind direction) [18,20,21,23,28,29]. Moreover, attention must be paid to the capability of trees to take up and transport different pollutants into their stem by comparing leaf-to-root and root-to-leaf pathways [30].

The use of dendrochemical techniques has the potential to define the retrospective biomonitoring of pollutants, and, in this context, advances are obtained by study areas well known to support the approach [31]. The Monte Amiata Mining District (MAMD; Central Italy) was chosen for the pollutant history of the study area. It was the third largest mercury (Hg) district worldwide (cumulative production of 102,000 tons; [32]), and currently hosts an important, actively exploited, geothermal field. Mining ceased in the early 1980s but left an impressive Hg contamination legacy in all environmental matrices, i.e., air, soil, sediments, water, and biota, as reported in recent years in several studies ([33] and references therein; [34–39]). Exploitation of geothermal energy in the area started in 1959; currently, there are five active plants equipped with emission control systems (AMIS) to reduce Hg and H$_2$S discharges from geothermal wells.

This research aims to prove the reliability of dendrochemical analysis to assess past and actual Hg pollution in the MAMD district, reconstructing a long-term record of Hg variations in an area affected by past Hg mining and metallurgical activities and the current exploitation of geothermal fluids. To the best of our knowledge, for the first time in this type of study, chestnut trees (*Castanea sativa* Mill.) were selected to test their ability to store a record of Hg changes in heartwood. Most published tree ring Hg records are developed at relatively coarse resolution (e.g., five-year resolution), and do not take full advantage of the annual resolution of the tree ring record [21]. Conversely, the tree ring analysis in this study provides a high-resolution (i.e., annual) temporal reconstruction of Hg exposure. Although currently Hg extraction, production, and commercialization are banned in most countries [40], it was a true "critical metal" for most of the XX century. The story presented here is, therefore, an instructive example of potential environmental concerns associated with the quest for critical resources.

## 2. Study Area

### 2.1. The Monte Amiata Area

The Monte Amiata area (Figure 1) (southern Tuscany, Italy) is characterized by a volcanic activity of dominant trachydacitic composition that occurred between 300 and 190 ka [41,42]. This volcanic activity caused the emplacement of a Pliocene magmatic

body at 6–7 km below the sea level. The anomalous heat flux related to this body likely triggered a hydrothermal system that caused the emplacement of Hg ore deposits [32,43,44]. Nowadays, the area hosts two geothermal reservoirs, which feed some $CO_2$-rich gas manifestations and Ca–$SO_4$-rich thermomineral waters, mainly located in NE and SW sectors of the volcanic edifice [45–47]. The Monte Amiata district is by extension the second largest geothermal field in Italy after Larderello.

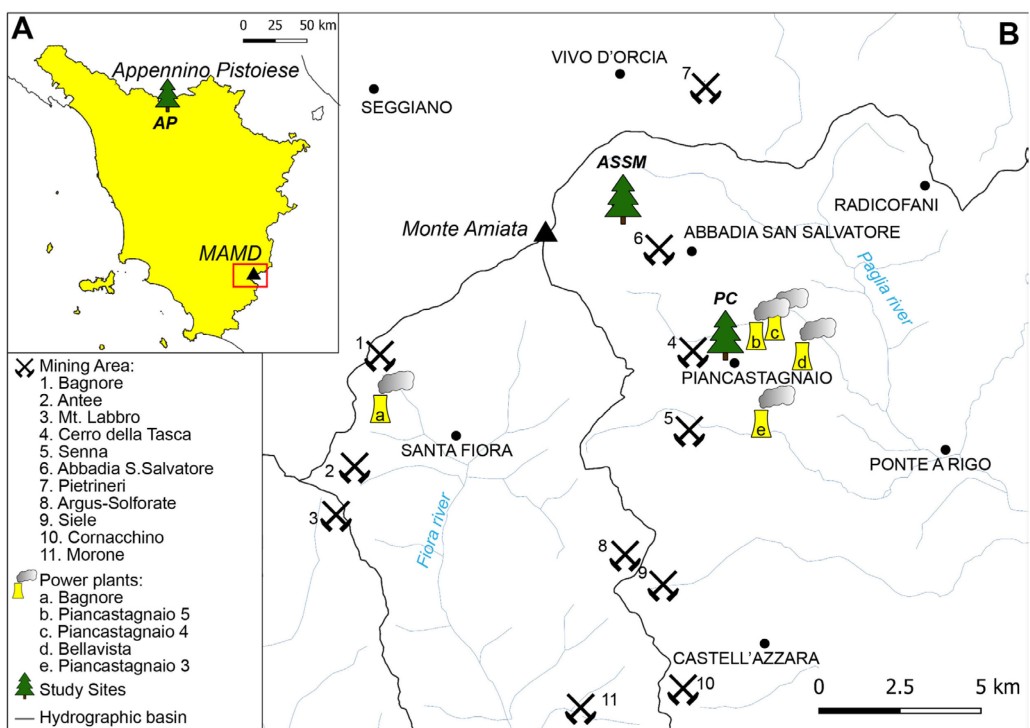

**Figure 1.** Study area and location of sampling sites. (**A**) Location of MAMD and AP sites, the red box indicates the position of Figure 2B. (**B**) The sampling location in the MAMD (ASSM and PC sites).

The climate of the area is temperate, with hot and dry summer, and cold rainy winter. In the last decade (2010–2020), the weather station of Abbadia San Salvatore-Laghetto Verde (TOS 11000114) reports an average annual temperature of 10.9 °C, and an average annual precipitation of 1543 mm (http://www.sir.toscana.it/consistenza-rete (accessed on 2 June 2022)). About two-thirds of the total annual precipitation are concentrated in the autumn–winter season [48]. Prevailing winds come from the W direction (280°), while subordinate directions are from NE (40°) and SE (140°) [38].

The Monte Amiata area is densely covered by a deciduous forest dominated by chestnut (*C. sativa* Mill.) below 1000 m, and by silver fir (*Abies alba* Mill.), beech (*Fagus sylvatica* L.), and black pine (*Pinus nigra* J.F. Arnold) above 1000 m [49]. Chestnut forests cover 7500 ha; half of the area (3500 ha) is managed for wood production under public ownership. The forests are located mainly in the eastern side of the region at 800 to 1200 m asl. Chestnut forests tend to be monospecific and are of anthropic origin; secondary species are rare or absent.

### 2.2. Mercury Sources and Mining Activity in the Monte Amiata Area

The main mining and smelting centers of the MAMD are mostly located in the central–eastern and southwestern sectors of the volcanic complex; the principal mining and metallurgical activity was in the municipality of Abbadia San Salvatore (Figure 1). The extraction and metallurgy of cinnabar (HgS) produced metallic Hg from the middle XIX century until the 1970s, when most of the Hg mines in the world were shut down due to environmental concerns.

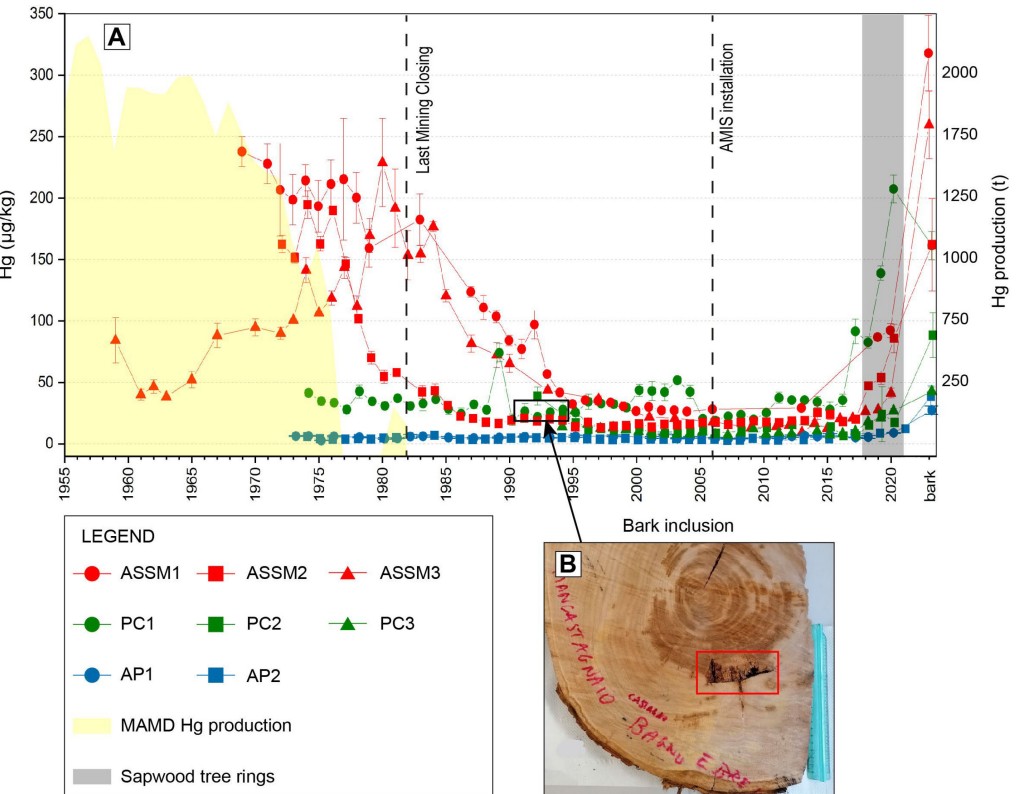

**Figure 2.** (**A**) Temporal Hg variation in the tree rings of ASSM (red lines), PC (green lines), and background area AP (blue lines). The yellow area represents the Hg production (tons) of the MAMD [50]. The grey area represents the sapwood tree rings. (**B**) Bark inclusion in PC1 sample in PC site (in the red box), dated 1991–1994.

Mining activity began in 1846–1847 with the exploitation of the deposits of Siele and Abbadia S. Salvatore, but, already by 1890, the production of Hg reached the remarkable value of about 450 t, i.e., 10% of the world production of that time [50]. One of the main production peaks occurred during the first decades of the 1900s, driven by the increasing demand of Hg fulminate employed during World War I. In 1917, the production of Hg had exceeded 1000 t, which corresponded to an impressive 26% of world production, overcoming Almadén in Hg flask trading [51]. After the economic crisis in 1930, production decreased and maintained low during World War II, since the district was heavily bombed. After the war, the Hg production had a new pulse during the Korean War up to the mid-1960s [51]; in the 1970s, the Hg demand began a constant decrease, down to a complete halt, with the consequent closure of the mines and plant production sites in 1982 [50]. Among the 42 mining sites and five metallurgical plants, the Abbadia San Salvatore mining area was the last to close its activities in 1981. Remediation of the mining sites occurred at Siele [35] and is ongoing at Abbadia San Salvatore [52].

The exploitation of geothermal energy at Monte Amiata started in 1959. In the study area, the first power plant (Bellavista; Figure 1) was built in 1987; currently, five power plants are active (Figure 1), with an installed capacity of 121 MWe [53]. The main environmental and health concerns of this activity arise from air emissions, represented by non-condensable gases (NCG) vented into the atmosphere, drift emitted from the cooling towers, noise emissions, and visual impact on the landscape. The main pollutants contained in the NCG emitted from geothermal power plants are hydrogen sulfide ($H_2S$) and Hg [54]. Mercury is present at very low concentrations in geothermal fluids, but its environmental mobility and high toxicity require abating the quantity emitted as much as possible [55]. For this reason, starting from 2002, all the geothermal plants currently operating in the Monte Amiata area are equipped with emission control systems (AMIS) to reduce Hg

and $H_2S$ emissions. The installation of AMIS in all the power plants was completed in 2007. However, the efficiency of these systems is not 100% and $Hg^0$ is still emitted from geothermal plants (from 1 to 4 g/h) [54].

### 2.3. Sampling Sites

Chestnut trees were sampled at two sites in the MAMD: (i) near the former Abbadia San Salvatore mine (ASSM), the largest mining and smelting center of the district and (ii) near the main geothermal power plant in the Piancastagnaio municipality (PC). At each site, three trees were sampled. Two trees were also sampled in the Appennino Pistoiese (AP; 150 km away from MAMD) that was selected as a reference background area (Figure 1).

## 3. Materials and Methods

### 3.1. Sampling and Sample Preparation

Taking profit of routine operations of forest maintenance, three tree disks (cross sections; ~5 cm in thickness) were collected from the stump at a height of 1 m at each sampling site (samples ASSM1, ASSM2, ASSM3 at ASSM; samples PC1, P2, PC3 at PC) in the Monte Amiata area, and two disks at the control site AP (samples AP1 and AP2). Disks were cut within one month from sampling using stainless-steel saws. A portion was polished with progressively finer sandpaper down to 600 grit to expose ring boundaries for identification and remove pollution retained on the rough surfaces. Stem discs allowed us to easily observe the stem profile, enabling easier separation of layers and providing adequate quantities for analysis. In the disk surface, sapwood is clearly recognized from heartwood because of its lighter color; sapwood typically comprises 3–4 growth rings.

After the surfaces of the tree disks were prepared, the tree rings were dated by counting back in time from the outermost ring, which represents the year of sampling (i.e., 2020). All tree rings were measured using a strip of graph paper under the binocular microscope. In addition, any exceptionally large ring or unique ring feature (e.g., impact damage) were noted as marker rings.

Individual disks were cut into one-year segments, starting with the most recent tree ring year, 2020, and moving toward the pith, using a stainless-steel knife; the selected fragments were air-dried. The one-year tree ring splitting was performed in nearly all samples, with the exception where low ring thicknesses made single splitting not possible. In the latter case, several tree rings were grouped and analyzed together: Table S1 in Supplementary Files report the detailed division of the trees. A clean scalpel blade was used to dissect the tree rings into annual growth increments and shave the outer surface to remove any potential superficial contamination that may have accumulated since the time of collection.

Water content was determined in a representative number ($n° = 30$) of dissected tree rings, following the procedure established by [38,56]. A sub-portion of approximately 1 g was dried in the oven at 110 °C for 24 h (or until a constant weight was achieved). The water content was calculated as a fraction of water in the total weight: water content (%) = [(wet weight) – (dry weight)]/(wet weight) × 100. All Hg measurements are reported on a dry weight basis. Water content is about 10% in sapwood, and 7% in heartwood.

The bark was also collected from each tree both at ASSM and PC, and dried. In sample PC1 collected at PC, a bark inclusion in tree stem was sampled and dated approximately between 1991 and 1994 (Figure 2B).

At each site, a representative soil sample was collected at a depth of 1–10 cm, removing the litterfall, i.e., the dead plant material (such as leaves, bark, needles, twigs, and cladodes) above the topsoil. Soils were sampled outside the tree canopy projection, within a distance of a few meters from the trunk. They were subsequently transported into the lab, air dried, sieved to select the fraction <2 mm, and ground.

*3.2. Chemical Analysis*

Total Hg concentrations in the tree rings sections representing the annual growth increments were quantified with a Milestone tri-cell Direct Mercury Analyzer (DMA-80) by atomic absorption, following thermal decomposition and amalgam preconcentration [57,58]. Tree rings were analyzed using the DMA-80 method recommended by the manufacturer for organic samples: 200 °C maximum starting temperature; 200 °C drying temperature; 50 s drying time; 800 °C decomposition temperature; 90 s decomposition time; and 60 s dwell time.

Soils were analyzed using the DMA-80 method for soil samples: 250 °C maximum starting temperature; 200 °C drying temperature; 40 s drying time; 650 °C decomposition temperature; 60 s decomposition time; and 60 s dwell time.

To estimate the analytical precision, three replicate analyses of the same sample (for both tree samples and soil samples) were routinely performed. Results were generally reproduced within <15% of the average value. Accuracy was evaluated using international standards (NIST 1575a, NIST 1573a, BCR-280R) and was within 10%. Concentrations are reported in µg/kg dry weight for tree rings and soils.

## 4. Results

The ages of sampled chestnut trees varied by study site (Table 1). At ASSM, trees were mainly 53 years old, ranging between 48 to 62 years, showing 3.5 mm per year as a mean tree ring increment. At PC, trees were younger than ASSM, ranging between 26 to 46 years with an average of 33 years, with a mean incremental growth of 4 mm per year.

**Table 1.** Characteristics of the study sites. Hg values are reported in µg/kg as average with standard deviation; the min and max values are reported within brackets.

| | Abbadia San Salvatore (ASSM) | Piancastagnaio (PC) | Appennino Pistoiese (AP) |
|---|---|---|---|
| Latitude | 42.89633 | 42.85409 | 44.04356 |
| Longitude | 11.64965 | 11.69055 | 10.76411 |
| Altitude (m asl) | 1070 | 725 | 438 |
| Tree species—sample name | *C. sativa* Mill.–ASSM1, ASSM2, ASSM3 | *C. sativa* Mill.–PC1, PC2, PC3 | *C. sativa* Mill.–AP1, AP2 |
| Tree age average (years) | 53 (48–62) | 33 (26–46) | 46–47 |
| Mean tree ring increments (mm) | 3.5 | 4 | - |
| Exposition | E | E | NW |
| Forest type | Deciduous forest | Mixed coniferous and deciduous forest | Mixed coniferous and deciduous forest |
| Hg source | Past mining activity | Geothermal power plant | Global atmospheric pollution |
| Distance from source (km) | 1.5 | 1 | - |
| Bedrock | Volcanic and volcano-sedimentary successions | Pliocene marine sediments | Flysch |
| Sample $n°$ | 3 | 3 | 2 |
| Hg in tree rings | 71.2 ± 65.8 (9.3–236.9) | 26.5 ± 26.8 (6.5–207.2) | 4.6 ± 2.1 (1.8–10.9) |
| Hg in tree rings before 1982 | 141.2 ± 61.2 (38.2–236.9) | 34.6 ± 5 (27.2–41.9) | 4.4 ± 1.6 (2.5–7) |
| Hg in tree rings before AMIS installation (2006) | 83.9 ± 69.1 (12.4–236.9) | 24.1 ± 13.3 (7.7–73.4) | 4.3 ± 1.8 (2.2–9.4) |
| Hg in bark | 245.7 ± 78.7 (161.1–316.9) | 97.5 ± 59.3 (43.5–161) | 32.8 ± 6.8 (27.9–37.6) |
| Hg in soil (µg/kg) | 3628 ± 2192 (1753–6038) | 1730 ± 742 (923–2383) | 117 ± 2 (116–119) |

In ASSM, Hg in tree rings varied from 9.3 to 236.9 µg/kg. In the mining period (until 1980) Hg concentrations in trees varied between 38.2 and 236.9 µg/kg, whereas in the post-production period after mine closure in the early 1980s, Hg showed a distinct decrease (between 9.3 and 176.7 µg/kg; Figure 2). It is to be noticed that the trend of Hg concentrations in tree rings does not exactly mimic the production trend (yellow area in Figure 2A). In particular, the Hg decreasing trend in trees begins with a slight delay (after 2 years) and extends for some years before reaching an almost constant plateau (post-1994). This fact is not surprising, considering that marked anomalies of atmospheric Hg are to this day recorded in proximity of smelting plants [34,52]. It is noted that samples ASSM1 and ASSM3 recorded a peak in 1980 (181.6 and 228.8 µg/kg, respectively) in correspondence of the last pulse of the Hg production of ASSM.

At PC, tree rings close to the geothermal plants (PC1, PC2, PC3) almost systematically showed a lower Hg content with respect to trees in the mining area (ASSM1, ASSM2, ASSM3), with an average of 26.5 ± 26.8 (average ± SD) µg/kg (ranging from 6.5 to 207.2 µg/kg). No significant variations were found in tree rings corresponding to the years after AMIS installation (early 2000s).

In the reference area (AP), the Hg concentrations varied from 1.8 to 10.9 µg/kg, indicating that trees were exposed to a modest local anomaly ([59] report 0.9 µg/kg for a reference site of a rural area).

All samples showed an increase in Hg concentration from heartwood to sapwood (up to an order of magnitude). Chestnut barks systematically showed higher Hg concentrations than sapwood (up to 316.9 µg/kg in ASSM; Figure 2A). The average Hg content in barks of ASSM is 245.7 ± 78.7 µg/kg, ranging between 161.1 and 316.9 µg/kg. In PC, barks showed an average value of 97.5 ± 59.3, ranging between 43.5 and 161 µg/kg. The bark inclusion in tree stem in the PC1 sample shows a content of 90 µg/kg. The background value in AP was 32.8 ± 6.8 µg/kg.

Soil samples collected at ASSM and PC showed concentrations of 3628 ± 2192 µg/kg and 1730 ± 74 µg/kg, respectively. These results are consistent with previous work on the presence of geochemical anomalies in the MAMD [32,60]. In the soils of the background area (AP site), Hg varied from 117 ± 2 µg/kg.

## 5. Discussion

Studies analyzing past Hg or other potentially toxic elements pollution recorded in tree rings are rare in Italy [31,61–64] and absent in the MAMD area. In this study, long-term reconstruction with an annual resolution of atmospheric $Hg^0$ in the MAMD district was achieved using chestnut (*C. sativa* Mill.) tree rings from 1958 to 2020.

The primary assumption to perform a dendrochemical analysis is the concept that each annual tree ring should reflect the trace elements available in the environment [65] and be absorbed by the tree through roots, leaves, and barks [30]. However, the pollutants absorption and storage in wood tissues is dependent by species; for instance, some authors ([8] and reference therein) observed in *Pinus* spp. that Hg concentration accumulated in tree rings fails to reconstruct the temporal trend of Hg production, in particular when coniferous species are employed, due to confounded tree physiological and environmental factors (i.e., the radial translocation and tree age effects occurring during the fast growing period).

However, the results presented in this study are broadly consistent with the basic dendrochemical assumption. The changes of Hg concentrations recorded in chestnut tree rings were temporally coherent with the closing of mining activities in the area (i.e., 1982) that would have altered Hg atmospheric concentrations (Figure 2).

On the other hand, although [66] have modeled that the AMIS installation has reduced the emission of $Hg^0$ by four times in the Piancastagnaio area, this variation has not been registered in our samples.

To the best of our knowledge, this is the first study in relation to tree rings collected in the MAMD which provides a temporal reconstruction of Hg exposure at annual resolutions. Results proved to be a good record of Hg deposition in areas affected by ore mining and smelting [22]. The reliability of the data is probably related to the object of investigation (i.e., Hg) and likely to the atmospheric pollutant that can be best traced by dendrochemical analyses [21,22,67], as well as to the employment of *C. sativa*, which is potentially the ideal arboreal species to track past Hg environmental concentrations.

Mercury concentrations found in the vascular tissues of woody plants (≅90%) are a result of $Hg^0$ stomatal and cuticular uptake by leaves. After its oxidation to divalent $Hg^{2+}$, it is incorporated in epidermal and stomatal cell walls [68], while the downward transport into tree rings occurs through the phloem and lateral translocation into the xylem [18,24]. Therefore, Hg concentrations found in tree rings could be conceivably considered as an evidence of past atmospheric Hg pollution. Other possible ways for Hg uptake by tree and translocation into woody tissues are Hg uptake from soil and movements from the outer bark into tree rings. However, both these mechanisms were suggested to be negligible [18,56,58,69,70]. In

the present study, the highest Hg concentrations in soil were found at ASSM, where Hg concentrations in wood are the highest. We cannot exclude the fact that root uptake from the soil is a source of Hg to the bole. However, the main species of total Hg in these soils is HgS ([32] and reference therein), a stable and very scarcely soluble Hg phase. Indeed, the translocation factor ($Hg_{wood}/Hg_{soil}$) is very low (0.019 ASSM; 0.015 PC; 0.039 AP). Thus, a soil Hg uptake seems unlikely, or in any case, minor.

Chestnut tree species are characterized by a relatively low and consistent number of sapwood tree rings able to record changes in atmospheric Hg concentrations more reliably than species with a relatively high and variable number of sapwood tree rings [8]. As previously stated, several authors [59,70,71] highlight that not all of the tree species are suitable for dendrochemical studies, because they clearly do not preserve Hg records. For example, some coniferous trees (e.g., *Pinus, Larix*), due to their permeable heartwood and the high number of rings in sapwood, may radially translocate Hg and other contaminants to older tree rings, smearing or otherwise altering Hg time series captured by tree rings [18]. By contrast, the nearly simultaneous response recorded in the collected chestnut (such as the peak observed by ASSM1 e ASSM3 in 1981 connected to the last pulse of Hg production of ASSM) trees indicated little or no translocation of Hg within the tree stem, which is in agreement with other works conducted on other tree species worldwide (e.g., larch in [70]). Among the species of the Fagaceae family and compared to other angiosperms, *C. sativa* wood has the highest concentrations of ellagitannins, a class of hydrolysable tannins, i.e., polyphenol compounds responsible for $Hg^{2+}$ sequestration in barks and involved in several plant reactions, such as redox transformations and cation complexation [72,73]. These compounds generally reach the highest level at the sapwood–heartwood boundary, and after heartwood formation, the no longer active cells become unable to accumulate these compounds [8,74]. In our opinion, the complexation of Hg with these compounds most likely results in the absence of Hg translocation between tree heartwood rings. This evidence is further confirmed by the bark inclusion of the PC1 chestnut (Figure 2B). The concentration of Hg (90 µg/kg) recorded in this bark, dated between 1991 and 1994, exactly followed the Hg positive spike (~73 µg/g) recorded in the previous years (1989) in this tree. No comparable Hg values were recorded in the adjacent rings, underlying the absence of translocation. Moreover, the physiology of heartwood formation could explain the increase in Hg concentration from heartwood to sapwood (up to an order of magnitude), and Hg concentration increase at the sapwood–heartwood boundary (Figure 2A).

Unlike heartwood, in sapwood living cells, in particular in the parenchyma ray cells, tannins are in solution; with the death of the cell, and so at the heartwood formation, tannins solidify and harden in the dead timber [75]. Therefore, radial permeability is enhanced in sapwood and may contribute in reducing temporal precision in the Hg concentrations measured in these rings. Translocation of divalent cations driven by living ray cells has, for example, been documented in oak sapwood [76,77]. Similarly, Hg concentrations found in chestnut trees could be considered not indicative of a single-year exposition, but may be influenced by the radial translocation.

Finally, chestnut barks systematically showed higher Hg concentrations than sapwood (Figure 2A, gray area), as already evidenced in previous studies both for Hg [8,78,79] and other heavy metals [80,81]. Mercury concentrations in barks layers (i.e., inner and outer barks) result from a differential Hg caption and absorption. Arnold et al. [18] and Peckham et al. [82] suggest that Hg translocation from the phloem may be an important source of Hg stored into the inner bark, while the Hg concentration in an outer bark mainly reflects atmospheric Hg depositions, both as gaseous and particulate Hg [56,83]. In this study, no distinction has been made between inner and outer chestnut barks, so Hg concentrations could be assumed as the sum of both bark layers and thus of Hg translocation and deposition.

## 6. Conclusions

Trunk cross-sections of 26 to 62 year old chestnut trees recorded in their growth rings the yearly evolution of atmospheric Hg in the Monte Amiata mining district and geothermal area (Italy). The oldest trees overlap with the last years of mining activity (1959–1982) and recorded a sharp decrease in Hg content in the rings following mine closure (after 1982). However, Hg contents in tree rings at Monte Amiata remain higher than in the control area 150 km away. This study showed that chestnut tree rings may represent a good record of atmospheric Hg in areas affected by past mining activities and geothermal power plants, and can be used as a low-cost, long-term biomonitoring tool for impact minimization, and the optimal management of environmental and land resources. However, the best results are achieved by selecting tree species with characteristically narrow sapwood and a consistent, low number of sapwood tree rings, such as sweet chestnut.

Moreover, the pathways of Hg uptake and translocation remain uncertain and difficult to study, especially in field settings, as in the MAMD. Further work is needed to assess the radial Hg translocation in more controlled studies with larger sample sizes and thus to satisfactorily interpret this historical Hg record.

**Supplementary Materials:** The following supporting information can be downloaded at: https://www.mdpi.com/article/10.3390/min13050688/s1, Table S1: Mercury concentration in chestnut tree rings.

**Author Contributions:** Conceptualization, P.C., S.F., F.C. and V.R.; methodology, S.F., F.C., G.M., C.C. and V.R.; validation, S.F. and F.C.; investigation, P.C., S.F., F.C. and V.R.; data curation, S.F., F.C. and V.R.; writing—original draft preparation, S.F. and F.C.; writing—review and editing, P.C., S.F., F.C., V.R., P.L., G.M., A.N., M.F. and C.C.; visualization, S.F.; supervision, P.C. and P.L. All authors have read and agreed to the published version of the manuscript.

**Funding:** This research received no external funding.

**Data Availability Statement:** Not applicable.

**Acknowledgments:** The authors would like to thank Sauro Visconti and all the staff of the *Unione dei Comuni Amiata Val d'Orcia* for their support during the sampling activity.

**Conflicts of Interest:** The authors declare no conflict of interest.

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
