# Peer review of "Tree Rings Record of Long-Term Atmospheric Hg Pollution in the Monte Amiata Mining District (Central Italy): Lessons from the Past for a Better Future"

_minerals, doi:10.3390/min13050688_

Round 1

Reviewer 1 Report

Paper by Fornasaro et al. is definitely nice case study with a dataset worth of publishing. Although I found some points where this work should be improved and strengthen.By the way it is the first study that includes a record of Hg in tree rings of deciduous tree making much sense! Maybe this could be pointed out little more.

First of all, there is NOT a definition of how the sapwood was determined in the methods? You only state that it was easy… Was it variable within the individual trees? How many tree rings it was formed by on the average and range?

Second of all, it is very suspective to see relatively flat line of tree ring Hg at the background sites. We know that atmospheric Hg (GEM) levels changed over time. They were elevated in the 1960-70s but the blue lines in figure 2 look quite flat. Paragraph in discussion would be good to deal with this. Speaking of figure 2 – It would it be much more comfortable for the reader if the individual sites would be named according to the area ASSM 1, ASSM 2… PC1, PC2 etc.

At two places (page 6, page 9) you use word “hardwood” instead of “heartwood” please check and correct.

Minor points to be corrected:

Abstract

Add word emissions and change verb:

                An additional source of Hg in the area is… An additional source of Hg emissions in the area have been active geothermal power plants.

Tree rings have little to do with deposition (directly):

                tree-rings may be a good record of Hg deposition in areas affected by… change to: tree-rings may be a good record of the atmospheric Hg changes in areas affected by…

Introduction

…Hg contamination over the long-term are lake cores, ice cores, stalagmites, and tree rings [14-17].                          Which work from the references in this sentence is about stalagmites?

                …dendrochemical analysis of tree rings proved to be an adequate tool to record atmospheric concentrations of Hg0 [18,19].                  Cooke et al. is a review paper. You should list here the real field studies such as Wright et al. 2014, Navrátil et al. 2018, Ghotra et al. 2020, Clackett et al. 2021, Nováková et al. 2021 etc.                          and all these references are already in your list!

                …were selected to test their capacity of Hg trapping and sequestration into the heartwood.          Maybe it would be better to say: …to test their ability to store a record of Hg changes in heartwood. Consider rewording, this way it feels like they capture a lot of mercury and store it in the heartwood which is not true.

Lines 118-124                    this is irrelevant to the study or please explain how is it relevant. I am wondering if thinning removes some of the trees completely or just some branches etc. Does any of this have an effect on Hg records?

Line 195                               should be (n=30)?

Paragraph 195-198           where are these data presented? Why did you determine the water content when the disks were air-dried for a 15 days as you say two paragraphs above?                    This is unclear and confusing…

Line 241 …average of 26.5 ± 26.8 (1SD?) μg/kg…                What does the question mark mean?

Line 255-256      … In the background area Hg varied from 117 ± 2 μg/kg.                 Do you mean Hg in the soil? If yes, say so.

Line 266               there is more works than this please add:

Leonelli, G., Battipaglia, G., Cherubini, P., Morra di Cella, U., & Pelfini, M. (2011). Chemical elements and heavy metals in European larch tree rings from remote and polluted sites in the European Alps. Geogr Fis Dinam Quat, 34, 195-206.

Orlandi, M., Pelfini, M., Pavan, M., Santilli, M., & Colombini, M. P. (2002). Heavy metals variations in some conifers in Valle d'Aosta (Western Italian Alps) from 1930 to 2000. Microchemical Journal, 73(1-2), 237-244.

Monticelli, D., Di Iorio, A., Ciceri, E., Castelletti, A., & Dossi, C. (2009). Tree ring microanalysis by LA–ICP–MS for environmental monitoring: validation or refutation? Two case histories. Microchimica Acta, 164, 139-148.

Line 272                this statement is about a certain species of trees (Pinus), you are generalizing and that is not good here. Please correct and make this more precise.

Line 282-284       this is a repetition, you said exactly this in the introduction Line85-87.

Line 300-301       you could at least try, what is the ratio between Hg in soil at the contaminated and reference areas? What is the ratio of Hg in the tree rings? The difference indicates that the root uptake is negligible, if any…

Line 308                it is not true that all the conifers have permeable heartwood.

Line 306-307      again this sound like these works have doubted tree ring Hg records in general BUT this is not true they only indicated that some species do not preserve the Hg record. Please rworite this accordingly.       (you should also include Siwik et al 2010).

Line 314-321       Great!

Line 323                this is new observation, how does the bark get in between the tree rings… this could be confusing for some readers and it is worth of more detailed explanation.

Line 339               this was also evidenced by , Navrátil et al. 2017 STOTEN and Nováková et al. 2021 STOTEN, add these references.

No comments, some small corrections could be made.

Author Response

Reviewer 1

Paper by Fornasaro et al. is definitely nice case study with a dataset worth of publishing. Although I found some points where this work should be improved and strengthen. By the way, it is the first study that includes a record of Hg in tree rings of deciduous tree making much sense! Maybe this could be pointed out little more.

First of all, there is NOT a definition of how the sapwood was determined in the methods? You only state that it was easy… Was it variable within the individual trees? How many tree rings it was formed by on the average and range?

The sapwood/heartwood distinction was made based on a clear color change (heartwood darker than sapwood). The distinction is always very sharp and clear in all samples; sapwood is typically formed by three to four growth rings. Sapwood is characterized by higher moisture content than heartwood (e.g. Taylor A.M., Gartner B.L., Morrel J.J. 2002. Heartwood formation and natural durability - A review. Wood and fiber science, vol. 34); indeed, we found about 10% water content in the sapwood, and about 7% in the heartwood (line 212).

Second of all, it is very suspective to see relatively flat line of tree ring Hg at the background sites. We know that atmospheric Hg (GEM) levels changed over time. They were elevated in the 1960-70s but the blue lines in figure 2 look quite flat. Paragraph in discussion would be good to deal with this.

An interesting comment. We point out some concepts 1. The flat trend is partly an artifact of the graph scale, that was conceived to demonstrate the much larger fluctuations at the Amiata sites. If you plot the AP data expanding the vertical scale, some fluctuations become evident. 2. The AP samples cover only few years of the 1960s-1970s; perhaps in that area fairly distant from significant point sources the effect of global scale variations is less perceived. 3. In any case, data from AP were collected for comparison purpose to demonstrate the strong anomaly at the Amiata sites. A detailed reconstruction of mercury budget in AP is outside the scope of this study.

Speaking of figure 2 – It would it be much more comfortable for the reader if the individual sites would be named according to the area ASSM 1, ASSM 2… PC1, PC2 etc.

We changed the sample names according to the suggestion.

At two places (page 6, page 9) you use word “hardwood” instead of “heartwood” please check and correct.

We corrected the mistakes.

Minor points to be corrected:

Abstract

Add word emissions and change verb:

An additional source of Hg in the area is… An additional source of Hg emissions in the area have been active geothermal power plants.

We corrected according to the suggestion.

Tree rings have little to do with deposition (directly):

Tree-rings may be a good record of Hg deposition in areas affected by… change to: tree-rings may be a good record of the atmospheric Hg changes in areas affected by…

We corrected according to the suggestion.

Introduction

…Hg contamination over the long-term are lake cores, ice cores, stalagmites, and tree rings [14-17]. Which work from the references in this sentence is about stalagmites?

We have eliminated “stalagmites” from the text. To the best of our knowledge, there are no specific case studies of mercury, only other trace elements; so, we prefer to cancel “stalagmites”.

…dendrochemical analysis of tree rings proved to be an adequate tool to record atmospheric concentrations of Hg0 [18,19].

Cooke et al. is a review paper. You should list here the real field studies such as Wright et al. 2014, Navrátil et al. 2018, Ghotra et al. 2020, Clackett et al. 2021, Nováková et al. 2021 etc. and all these references are already in your list!

Added and corrected the references in the text.

…were selected to test their capacity of Hg trapping and sequestration into the heartwood.

Maybe it would be better to say: …to test their ability to store a record of Hg changes in heartwood. Consider rewording, this way it feels like they capture a lot of mercury and store it in the heartwood which is not true.

We rewrote the sentence according to the suggestion.

Lines 118-124 this is irrelevant to the study or please explain how is it relevant. I am wondering if thinning removes some of the trees completely or just some branches etc. Does any of this have an effect on Hg records?

You are right, this part is not relevant to the study. We have decided to eliminate it.

Line 195 should be (n=30)?

Corrected according to the suggestion.

Paragraph 195-198 where are these data presented? Why did you determine the water content when the disks were air-dried for a 15 days as you say two paragraphs above? This is unclear and confusing…

We corrected the text at line 182 and 202-206. Water was determined after tree rings cutting, which occurred within 1 month from sampling. Water was determined in a sub-portion of tree disks (not further analyzed for Hg) following a procedure established in our previous works (Chiarantini et al. 2016; Rimondi et al., 2020). We point out that 15 days of natural seasoning aren’t enough for a homogeneous distribution of moisture inside the wood, given the thickness of the disks. The determination of the actual moisture content (MC) of each sample was important because MC affects any wood properties.

Line 241 …average of 26.5 ± 26.8 (1SD?) μg/kg…

What does the question mark mean?

It was a typing error, we corrected it.

Line 255-256… In the background area Hg varied from 117 ± 2 μg/kg.

Do you mean Hg in the soil? If yes, say so.

We corrected according to the suggestion.

Line 266 there is more works than this please add:

Leonelli, G., Battipaglia, G., Cherubini, P., Morra di Cella, U., & Pelfini, M. (2011). Chemical elements and heavy metals in European larch tree rings from remote and polluted sites in the European Alps. Geogr Fis Dinam Quat, 34, 195-206.

Orlandi, M., Pelfini, M., Pavan, M., Santilli, M., & Colombini, M. P. (2002). Heavy metals variations in some conifers in Valle d'Aosta (Western Italian Alps) from 1930 to 2000. Microchemical Journal, 73(1-2), 237-244.

Monticelli, D., Di Iorio, A., Ciceri, E., Castelletti, A., & Dossi, C. (2009). Tree ring microanalysis by LA–ICP–MS for environmental monitoring: validation or refutation? Two case histories. Microchimica Acta, 164, 139-148.

We add these references in the text.

Line 272 this statement is about a certain species of trees (Pinus), you are generalizing and that is not good here. Please correct and make this more precise.

We rewrote the sentence.

Line 282-284 this is a repetition, you said exactly this in the introduction Line 85-87.

We rewrote the sentence.

Line 300-301 you could at least try, what is the ratio between Hg in soil at the contaminated and reference areas? What is the ratio of Hg in the tree rings? The difference indicates that the root uptake is negligible, if any…

Poor Hg translocation is documented by a lot of studies (Arnold et al., 2018; Chiarantini et al., 2016; Bishop et al., 1998; Navratil et al., 2018), as also indicated in the text. The intent of our paper was not to further test this evidence. In any case, following the reviewer suggestion, we report here the translocation factors (Hgwood/Hgsoil) for our sample sites (0.019 ASSM; 0.015 PC; 0.039 AP).
These factors are in fact very low, supporting the idea that uptake from the roots is negligible.

Line 308 it is not true that all the conifers have permeable heartwood.

We corrected the sentence.

Line 306-307 again this sound like these works have doubted tree ring Hg records in general BUT this is not true they only indicated that some species do not preserve the Hg record. Please wrote this accordingly (You should also include Siwik et al 2010).

We rewrote the sentence to clarify this concept.

Line 314-321Great!

Thanks!

Line 323 this is new observation, how does the bark get in between the tree rings… this could be confusing for some readers, and it is worth of more detailed explanation.

We add more info in the results section.

Line 339 this was also evidenced by Navrátil et al. 2017 STOTEN and Nováková et al. 2021 STOTEN, add these references.

We add these references in the text.

Reviewer 2 Report

This paper from Fornasaro et al. provides new insight into atmospheric Hg pollution in a mining district in central Italy using tree rings for long-term records. The paper is generally well-written with sound methodology.  The author's data analysis and interpretation are good. The paper is also of sufficient quality and quantity. In my opinion, it is worthy of publication. Below are some minor comments that should, I think, improve the paper.

Consider removing “for a better future” in the title.  I think “Lessons from the past” is enough.

Line 24: Consider adding “likely” in front of “because of mine closure” in the abstract.   

Section 3.2: Consider adding references for the use of the direct mercury analyzer based on thermal decomposition atomic absorption spectrometry in prior studies of Hg in tree rings:

Molecules 202025(9), 2126; https://doi.org/10.3390/molecules25092126

Water Air Soil Pollut 231, 248 (2020). https://doi.org/10.1007/s11270-020-04601-2

Also in Section 3.2: Discuss how the samples were dried for determining the dry weight concentrations.

Author Response

This paper from Fornasaro et al. provides new insight into atmospheric Hg pollution in a mining district in central Italy using tree rings for long-term records. The paper is generally well-written with sound methodology. The author's data analysis and interpretation are good. The paper is also of sufficient quality and quantity. In my opinion, it is worthy of publication. Below are some minor comments that should, I think, improve the paper.

Consider removing “for a better future” in the title. I think “Lessons from the past” is enough.

Thanks for the suggestion, however we prefer to keep the original title.

Line 24: Consider adding “likely” in front of “because of mine closure” in the abstract.

We added it.

Section 3.2: Consider adding references for the use of the direct mercury analyzer based on thermal decomposition atomic absorption spectrometry in prior studies of Hg in tree rings:

Molecules 2020, 25(9), 2126; https://doi.org/10.3390/molecules25092126

Water Air Soil Pollut 231, 248 (2020). https://doi.org/10.1007/s11270-020-04601-2

We added these references in the text.

Also in Section 3.2: Discuss how the samples were dried for determining the dry weight concentrations.

We modified the Section according to the suggestion of the reviewers.

Reviewer 3 Report

The presented study is devoted to a very important topic - the study of environmental pollution by a substance of the first hazard class - mercury. On the territory of a former mining enterprise, annual rings of chestnut trees were studied, and an assessment was made of the possibility of using this method for the purposes of the chronology of changes in the mercury content in the atmosphere of the study area.

The work is complex, interesting, has scientific and applied value.

In general, I think that the work can be published in the form presented, there were very few comments.

166-167 Figure 1.

It is not entirely clear how the two cards relate. So that there are no questions, you can indicate with an arrow where the part of the map on the right is taken and enlarged, where most of the tested trees are located.

211 "Tree rings were analyzed using the DMA-80 method recommended by the manufacturer for organic samples."

If possible, you need to refer to the methodology (where you can get acquainted with it - perhaps there are already works where the method is described).

Author Response

The presented study is devoted to a very important topic - the study of environmental pollution by a substance of the first hazard class - mercury. On the territory of a former mining enterprise, annual rings of chestnut trees were studied, and an assessment was made of the possibility of using this method for the purposes of the chronology of changes in the mercury content in the atmosphere of the study area.

The work is complex, interesting, has scientific and applied value.

In general, I think that the work can be published in the form presented, there were very few comments.

166-167 Figure 1.

It is not entirely clear how the two cards relate. So that there are no questions, you can indicate with an arrow where the part of the map on the right is taken and enlarged, where most of the tested trees are located.

We modified the figure according to the suggestion.

211 "Tree rings were analyzed using the DMA-80 method recommended by the manufacturer for organic samples".

If possible, you need to refer to the methodology (where you can get acquainted with it - perhaps there are already works where the method is described).

We added some references according to the suggestion of the reviewer #2.